# Numerical Analysis of an Explicit Smoothed Particle Finite Element Method on Shallow Vegetated Slope Stability with Different Root Architectures

Xichun Jia [1], Wei Zhang [1], Xinghan Wang [2], Yuhao Jin [1,*] and Peitong Cong [1,*]

[1] College of Water Conservancy and Civil Engineering, South China Agricultural University, Guangzhou 510642, China
[2] Water Resources Remote Sensing Department, Pearl River Water Resources Research Institute, Guangzhou 510610, China
* Correspondence: yuhao.jin@scau.edu.cn (Y.J.); slxyky@scau.edu.cn (P.C.)

**Abstract:** Planting vegetation is an environmentally friendly method for reducing landslides. Current vegetated slope analysis fails to consider the influence of different root architectures, and the accuracy and effectiveness of the numerical simulations need to be improved. In this study, an explicit smoothed particle finite element method (eSPFEM) was used to evaluate slope stability under the influence of vegetation roots. The Mohr–Coulomb constitutive model was extended by incorporating apparent root cohesion into the shear strength of the soil. The slope factors of safety (FOS) of four root architectures (uniform, triangular, parabolic, and exponential) for various planting distances, root depths, slope angles, and planting locations were calculated using the shear strength reduction technique with a kinetic energy-based criterion. The results indicated that the higher the planting density, the stronger the reinforcement effect of the roots on the slope. With increasing root depth, the FOS value first decreased and then increased. The FOS value decreased with an increase in slope angle. Planting on the entire ground surface had the best improvement effect on the slope stability, followed by planting vegetation with a uniform root architecture in the upper slope region or planting vegetation with triangular or exponential root architecture on the slope's toe. Our findings are expected to deepen our understanding of the contributions of different root architectures to vegetated slope protection and guide the selection of vegetation species and planting locations.

**Keywords:** explicit smooth particle finite element method; vegetated slope; root architectures; shear strength reduction technique; factor of safety

## 1. Introduction

Landslides and debris flows are common natural disasters that cause environmental damage, human casualties, and economic losses [1,2]. To tackle this problem, measures, such as nailing, vegetation, ground improvement, geosynthetic reinforcement, and improved drainage have been taken. Among these, vegetation is an economically, sustainable, and environmentally friendly bio-remediation technique [3,4]. Further studies on the effects of vegetation on slope stability are essential.

It is now understood that vegetation contributes to the stability of civil infrastructure, including shallow slopes (i.e., slopes with vertical depths less than 2 m), road and railway slopes, dams, embankments, and dykes [5–9]. Vegetation roots (e.g., from grasses, shrubs, and trees) are believed to stabilise slopes and slow the large-scale movement of landslides by strengthening soils with increased cohesion [10–13]. Planting trees on slopes can reduce the occurrence of shallow landslides by up to 95% compared with similar areas without trees [14,15].

To date, a large body of literature has documented studies that have focused on quantifying the contribution of roots to soil shear strength. These studies included in situ

direct shear tests [16,17] and laboratory direct shear tests [18,19] on soil blocks containing plant roots, as well as laboratory direct shear tests of soils reinforced by fibres that simulate roots [20–22]. The finite element method (FEM) can also be applied in this vein of research [23,24]. These studies have shown that roots can increase the shear strength of the soil. Therefore, it is necessary to incorporate the effects of root reinforcement into landslide prediction models and slope-stability analyses.

The limit equilibrium method (LEM) [25,26] and FEM [27,28] have been widely used in slope-stability analyses. The advantages of FEM compared to LEM are that it does not need to presuppose the shape and position of the critical slip surface, and the stress–strain relationship and soil deformation behaviour can be considered using the FEM. Slope-stability analysis based on the FEM has been recognised as an effective tool in geotechnical engineering [29]. The FEM has also been applied to the analysis of vegetated slopes [30]. The traditional FEM can correctly describe the initial failure surface, but for the large deformation problem of the slope after the initial failure, the numerical simulation may be inaccurate or even impossible owing to mesh distortion; therefore, FEM has limitations to such problems [29,31].

An alternative solution for overcoming the FEM mesh distortion problem is the meshless technique, which uses a set of particles to replace the mesh in FEM-based approaches [32]. Many meshless numerical frameworks have been presented to solve large-deformation problems in geomechanics so far [33], such as discrete element method (DEM) [34,35], smoothed particle hydrodynamics (SPH) [36,37], and the material point method (MPM) [38,39]. However, meshless methods require neighbours' searching, which needs high computation costs. Moreover, meshless methods usually necessitate special treatment techniques to deal with boundary conditions [32]. The particle finite element method (PFEM) [40,41] uses particles to represent materials that are similar to those in the mesh-free method, and it has been proven as a powerful numerical means to analyse the post-failure mechanisms in geo-engineering [32]. In recent years, many novel methods have been improved on the basis of PFEM, such as the smoothed particle finite element method (SPFEM) [30,32,42,43], edge-based smoothed PFEM (ES-PFEM) [44], node-based smoothed PFEM (NS-PFEM) [45], and stable node-based smoothed PFEM (SNS-PFEM) [46,47].

In previous studies, the numerical analysis of the influence of roots on slope stability usually involved the change of material parameters at a certain depth of the soil layer, and the effect of specific root architecture has seldom been considered. Plant roots penetrate the soil matrix to form a root-soil composite [29,31]. Nevertheless, existing analytical models generally concentrate on the ultimate limit state, and they neglect the complicated interactions between root systems and soil [48]. Therefore, quantifying the root contribution and determining the critical slip surface remains a challenge.

Based on the shortcomings of previous investigations, this study proposes a novel method for evaluating the stability of shallow vegetated slopes. An eSPFEM with a kinetic-energy-based criterion was used for the numerical simulation. The apparent root cohesion is incorporated into the Mohr–Coulomb constitutive model. The FOS of the four root architectures for various planting distances, root depths, slope angles, and planting locations were calculated using the shear strength reduction technique. To the best of our knowledge, this is the first study to use eSPFEM to calculate the effects of different root architectures on slope stability. This study is expected to provide a reference for improving the slope stability and optimising the management of mountain shelter forests. Nonetheless, the limitation of this study lies in the need for a large amount of calculation and accurate parameterisation, so it is currently only applicable to small and shallow slopes.

## 2. Materials and Methods

### 2.1. The eSPFEM Approaches

The PFEM solves the governing equations using a standard finite element approach [30]. Therefore, it not only has the flexibility of mesh-free particle methods for arbitrary changes in geometry but also inherits the solid mathematical foundation of the traditional FEM [49].

The PFEM is based on an updated Lagrangian (UL) fashion for modelling the motion of a continuum medium. The continuum medium is discretised into a set of Lagrangian nodes (particles) that contain and transmit all of the information present. The computational mesh was built using the Delaunay triangulation technique, and the boundary of the computational mesh was identified using the alpha-shape approach. A mesh was then used to solve the governing equations. However, in the PFEM, excessive mesh distortion is avoided by frequent remeshing, and the state variables (i.e., stress and strain) are mapped from old Gauss points to new ones. This mapping procedure inevitably introduces errors, which increase the complexity of the calculation process [30,32,42].

The SPFEM [50,51] uses a strain smoothing technique for nodal integration based on the PFEM to achieve the balance of governing equations at nodes or particles. In the SPFEM, all of the field variables are calculated at particles instead of Gauss points in the PFEM, which can avoid information transfer between Gauss points and particles, thus reducing the calculation error. In addition, the SPFEM possesses the upper-bound property and provides a conservative estimate for problems in geomechanics. Finally, low-order triangle elements can be used directly without volumetric locking [30,32,42]. The SPFEM can consider the entire dynamic failure process of a slope in slope-stability analysis and therefore simulate the large deformation and post-failure of soil to obtain a more reliable FOS value [32].

In SPFEM, the computation domain is discretized into strain smoothing cells associated with nodes. The physical volume, $\Omega$, is correspondingly discretized into particles. As shown in Figure 1, The smoothing cell associated with the particle $k$ is created by connecting sequentially the mid edge point to the central points of the surrounding triangular elements of particle $k$. A strain smoothing operation is then performed based on the set of smoothing cells that are created based on the triangulation mesh [32].

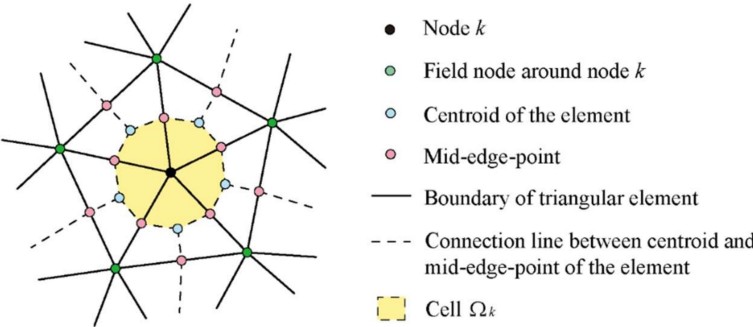

**Figure 1.** Construction of smoothing cell associated with particle $k$.

In the smoothed strain technique, the area, $A_k$, of the smoothing cell, $\Omega_k$, associated with node $k$ is calculated by:

$$A_k = \int_{\Omega_k} d\Omega = \frac{1}{3}\sum_{j=1}^{N_k} A_j \qquad (1)$$

where $N_k$ is the number of elements related to particle k and $A_j$ is the area of the $j$th element. The smoothed strain matrix is calculated by:

$$\widetilde{\boldsymbol{B}}_k = \frac{1}{A_k}\sum_{j=1}^{N_k}\frac{1}{3}A_j\boldsymbol{B}_j \qquad (2)$$

where $B_j$ represents the strain gradient matrix used in the standard FEM for the $j$th element. More details regarding the strain smoothing technique are available in Liu et al. [51] and Chen et al. [52].

The discretization of the computational formulations of eSPFEM is now briefly presented. The motion of a continuum can be described as:

$$\rho \boldsymbol{a} = \nabla \cdot \boldsymbol{\sigma} + \rho \boldsymbol{b} \tag{3}$$

where, $\rho$ is the material density, $\boldsymbol{a}$ is the acceleration, $\boldsymbol{\sigma}$ is the Cauchy stress tensor, and $\boldsymbol{b}$ is the specific body force density. Considering the principle of virtual displacement and the divergence theorem, the weak form is expressed as:

$$\int_{\boldsymbol{\Omega}} \delta \boldsymbol{u} \cdot \rho \boldsymbol{a} d\Omega = \int_{S} \delta \boldsymbol{u} \cdot \boldsymbol{\tau}_S dS + \int_{\boldsymbol{\Omega}} \delta \boldsymbol{u} \cdot \rho \boldsymbol{b} d\Omega - \int_{\boldsymbol{\Omega}} \delta \boldsymbol{u} : \boldsymbol{\sigma} d\Omega \tag{4}$$

where $\boldsymbol{u}$ is the displacement vector, $\Omega$ represents the configuration domain, $S$ represents the boundary, and $\boldsymbol{\tau}_S$ is the prescribed traction. After the node-based discretization, the above equation reads:

$$\sum_{k=1}^{T} \rho \boldsymbol{a}_k A_k = \sum_{k=1}^{T} \int_{S} \boldsymbol{N}_k \boldsymbol{\tau}_S dS + \sum_{k=1}^{T} \rho \boldsymbol{b}_k A_k - \sum_{k=1}^{T} \widetilde{\boldsymbol{B}}_k^{T} \boldsymbol{\sigma}_k A_k \tag{5}$$

where $T$ is the total number of nodes in the computation domain. The above discrete form can then be written in the vector or matrix form as:

$$\boldsymbol{M}\boldsymbol{a} = \boldsymbol{F}^{ext} - \boldsymbol{F}^{int} \tag{6}$$

in which:

$$\boldsymbol{F}^{ext} = \sum_{k=1}^{T} \int_{S} \boldsymbol{N}_k \boldsymbol{\tau}_S dS + \sum_{k=1}^{T} \rho \boldsymbol{b}_k A_k \tag{7}$$

$$\boldsymbol{F}^{int} = \sum_{k=1}^{T} \widetilde{\boldsymbol{B}}_k^{T} \boldsymbol{\sigma}_k A_k \tag{8}$$

$$\boldsymbol{M} = \sum_{k=1}^{T} \rho A_k \tag{9}$$

where $\boldsymbol{F}^{ext}$ and $\boldsymbol{F}^{int}$ are termed as the external and internal forces, respectively, and $\boldsymbol{M}$ is the diagonal mass matrix.

A typical computational cycle of eSPFEM is shown as follows [43]:

(1) Generate mesh using Delaunay triangulation and alpha shape technique;
(2) Acquire basic data of elements and nodes;
(3) Calculate smoothed strains of nodes;
(4) Update stresses of nodes through constitutive integration;
(5) Calculate internal forces of nodes;
(6) Update velocities and position.

Compared with the implicit SPFEM [30], the eSPFEM adopts an explicit time-integration scheme, which has a more concise formulation, lower computational cost, and wider applications. We used a self-developed code that aims to develop a GPU-accelerated SPFEM for large deformation analysis in geomechanics based on CUDA [43], which is released by NVIDIA to perform high-performance computing and has gained popularity rapidly in geomechanics recently [53,54]. Details of the eSPFEM theory can be found in Yuan et al. [32,42] and Zhang et al. [43,55,56].

### 2.2. Modelling the Mechanical Effect of Roots

Plant roots extend into the soil matrix and form a soil-root composite material with a mechanical effect that enhances the shear strength of the soil [57]. This effect is often considered as additional soil cohesion (known as apparent root cohesion) [23,58,59].

Different theoretical models have been created to estimate the mechanical effect of roots on slope stability, such as the Wu model [17,60] and the fibre bundle model (FBM) [61]. The Wu model assumes that all roots are moved and broken simultaneously, which is not true in reality. However, it is the most commonly used model owing to its simplicity. The Wu model is a perpendicular root-strengthening model established using two variables: root area ratio (*RAR*) and root tensile strength [48].

The Wu model is used to describe the increase in soil shear strength caused by the mechanical effect of the roots, as follows:

$$c_r = \zeta \times T_r \times R_f \times RAR \tag{10}$$

where $c_r$ is the additional soil cohesion and $\zeta$ represents the correction factor, which takes into account the influence of roots breaking progressively in reality on soil shear strength; in the present study, $\zeta$ equals 0.4 [62].

$T_r$ is the root tensile strength. For simplicity, the effects of the root diameter distributions were ignored, and a constant root tensile strength was considered [63]. Available current data on tree roots indicate that $T_r$ ranges from 5 to 60 MPa [11,64]; in the present study, $T_r$ equals 20 MPa.

$R_f$ stands for the root orientation factor, which is defined as follows:

$$R_f = \sin\theta + \cos\theta \tan\varphi' \tag{11}$$

where $\theta$ represents the angle between the root and failure surface when the root breaks, and $\varphi'$ is the effective angle of internal friction. It should be noted that roots do not always grow perpendicular to the slope's surface, as their growth is influenced by ambient conditions (e.g., gravity and nutrition) [57]. In many cases, $\theta$ is between 48° and 72° at failure, so the range of $\sin\theta + \cos\theta \tan\varphi'$ is narrow, i.e., approaching 1.2 [11]. In the present study, $R_f$ equals 1.2.

The *RAR* (in Equation (1)), which is defined as the proportion of the cross-sectional area of the soil occupied by roots, is determined as follows:

$$RAR = \frac{A_r}{A} \tag{12}$$

where $A_r$ and $A$ represent the cross-sectional area of root and soil, respectively. In the present study, *RAR* equals 0.45% [64].

Many researchers have estimated the value of $c_r$ for various vegetation species growing in different environments, and its typical values vary from 1.0 to 94.3 kPa [29]. According to Equation (1), the original $c_r$ score in this study is 43.2 kPa.

### 2.3. Mechanical Effect of Root-Soil on Shallow Slope Stability

Most of the results from the direct-shear tests showed that roots increased the cohesion [60], whereas the friction angle remained mostly unchanged [64]. For undrained loading, the shear strength of saturated vegetated soils, $s$, can commonly be determined by incorporating the additional soil cohesion, $c_r$, into the Mohr–Coulomb failure criterion [64,65], which can be modified as follows:

$$s = c_r + c' + \sigma_n \tan\varphi' \tag{13}$$

where, $c'$ is effective cohesion and $\sigma_n$ is total normal stress.

The shear strength reduction technique was adopted to analyse the slope stability, which is defined as follows:

$$\begin{aligned} s_f &= \tfrac{s}{\text{SRF}} \\ \varphi' f &= \arctan\left(\tfrac{\tan\varphi'}{\text{SRF}}\right) \end{aligned} \tag{14}$$

where $s_f$ and $\varphi'_f$ are the reduced shear strength and the reduced effective angle of internal friction, respectively. SRF is the shear strength reduction factor. First, the initial value of the SRF was set to an adequately low value to keep the slope stable under gravitational loading. Then, the value of the SRF increases gradually until the slope becomes unstable and failure occurs. The critical value of the SRF leading to slope instability is considered to be the FOS of the slope [30].

### 2.4. The Four Patterns of Root Geometry

#### 2.4.1. Idealisation of Typical Patterns of Root Architecture

The spatial location of thick roots determines the arrangement of related thin roots; thus, root distribution presents a high degree of complexity [8]. Existing methods for quantifying root architecture include the extraction of roots, the complete washing of soil, and image analysis of roots [66]. The spatial distribution of roots is an important factor in determining the reinforcement behaviour and mechanical properties of roots, and the generalisation of root morphology is essential for evaluating the influence of vegetation on slope stability [9,67].

Based on experimental observations, researchers have summarised four typical patterns of root geometry: uniform distribution [10,68], triangular distribution [68], parabolic distribution [69], and exponential distribution [10]. Table 1 lists the characteristics of different root architectures, typical species, and their growing regions. Figure 2 shows the root architecture of four different real roots.

**Table 1.** Introduction to different root architectures.

| Root Architectures | Characteristics | Typical Species | Growing Regions |
|---|---|---|---|
| Uniform distribution | A root system with a large taproot and large horizontal lateral roots | Aleppo pine [10] | Mediterranean, Southern Europe, Asia, and North Africa [70] |
| | | Pulsatilla pratensis [68] | Sub-polar areas of Europe, Asia, North America Central, and Eastern Europe [71] |
| Triangular distribution | A root taproot system with small lateral roots | Trigonella balansae [68] | Europe, and Asia [72] |
| Parabolic distribution | A concentrated root system | Cynodon dactylon [69] | North Africa, Asia, Australia, and Southern Europe [73] |
| Exponential distribution | A plate-shaped root system | Beech and Mature oaks [10] | Europe, and North America [74,75] |

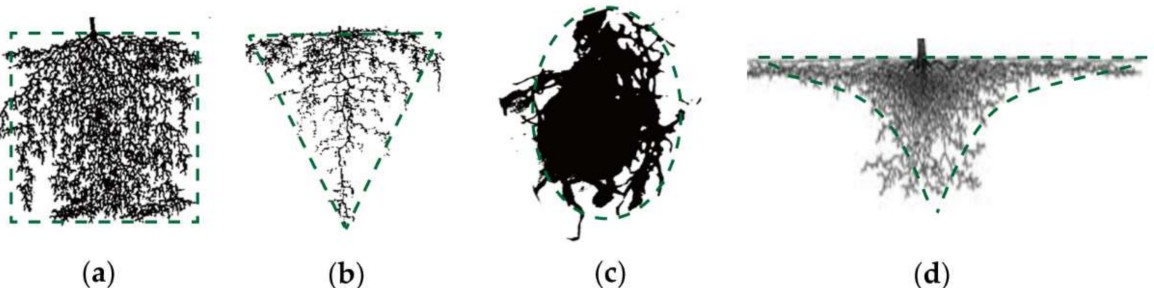

**Figure 2.** Different real root architectures: (**a**) Uniform distribution [68]; (**b**) Triangular distribution [68]; (**c**) Parabolic distribution [69]; (**d**) Exponential distribution [10].

Root biomass can be expressed by root volume, mass, area, or length, and the most commonly adopted measure is the soil area occupied by the roots [76]. In order to achieve a fair comparison, the symmetric parts of the four root architecture profiles are normalised to the same unit area. Figure 3 describes their normalised function curves. It is assumed

that these root architectures have the same root depth and are homogeneously distribute in the root zone.

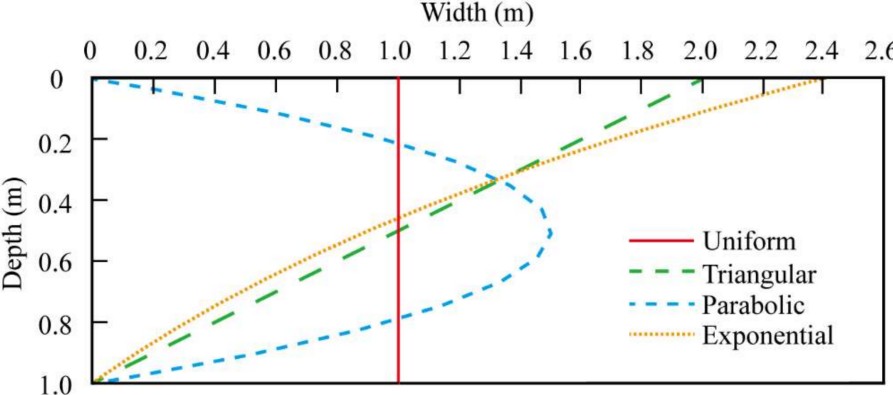

**Figure 3.** Normalised function curves of the four different root architectures.

### 2.4.2. Root Architecture Functions on Slope

The coordinate system on the slope can be defined as $x'o'y'$, and the extent between the bottom of the root zone and the slope's surface is defined by the root depth, $z_r$. Figure 4 describes the boundaries of the four different root zones, and the green areas indicate the root zones. The mathematical functions derived from the root zone boundaries are as follows:

1.  Uniform distribution

$$\begin{cases} y' = \frac{1}{z_r} \\ y' = -\frac{1}{z_r} \\ x' = 0 \\ x' = -z_r \end{cases} \tag{15}$$

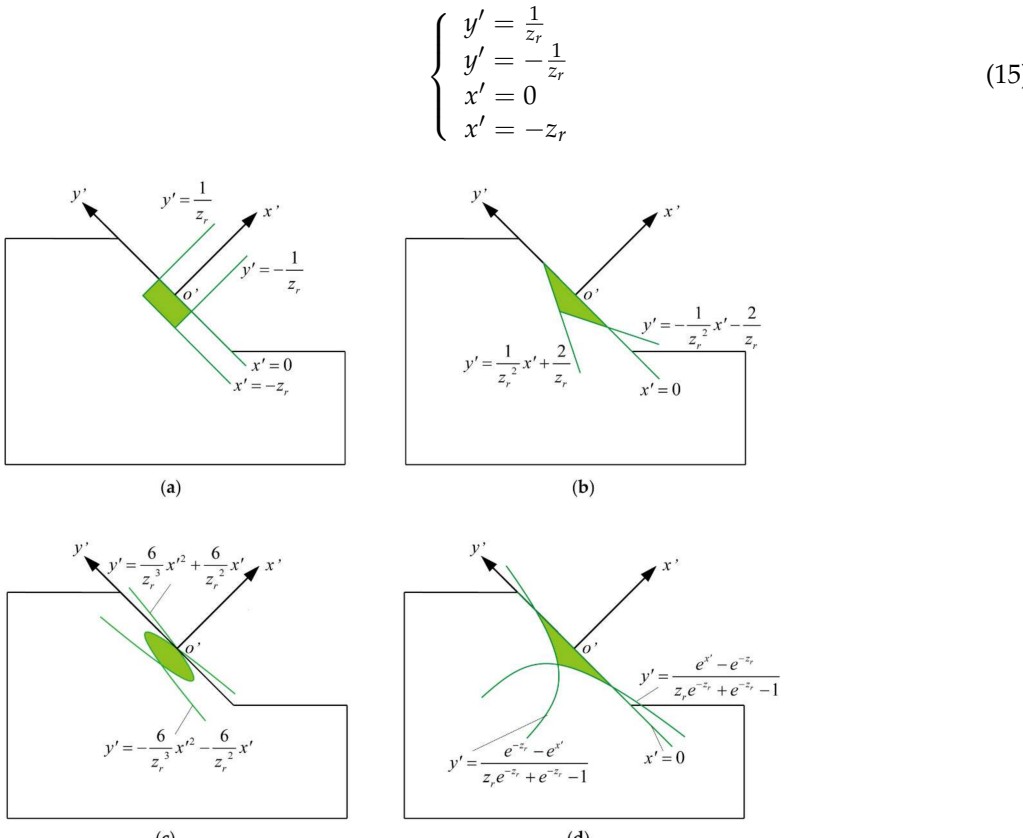

**Figure 4.** Boundaries of the four different root zones: (**a**) Uniform distribution; (**b**) Triangular distribution; (**c**) Parabolic distribution; (**d**) Exponential distribution.

2.  Triangular distribution

$$\begin{cases} y' = -\frac{1}{z_r{}^2} x' - \frac{2}{z_r} \\ y' = \frac{1}{z_r{}^2} x' + \frac{2}{z_r} \\ x' = 0 \end{cases} \tag{16}$$

3. Parabolic distribution

$$\begin{cases} y' = \frac{6}{z_r{}^3} x'^2 + \frac{6}{z_r{}^2} x' \\ y' = -\frac{6}{z_r{}^3} x'^2 - \frac{6}{z_r{}^2} x' \end{cases} \tag{17}$$

4. Exponential distribution

$$\begin{cases} y' = \frac{e^{x'} - e^{-z_r}}{z_r e^{-z_r} + e^{-z_r} - 1} \\ y' = \frac{e^{-z_r} - e^{x'}}{z_r e^{-z_r} + e^{-z_r} - 1} \\ x' = 0 \end{cases} \tag{18}$$

2.4.3. Root Architecture Functions after Coordinate Transformation

The coordinate transformation formula from the coordinate system $x'o'y'$ to the rectangular coordinate system $xoy$ is assembled as:

$$\begin{bmatrix} x \\ y \end{bmatrix} = \begin{bmatrix} \cos\alpha & -\sin\alpha \\ \sin\alpha & \cos\alpha \end{bmatrix} \begin{bmatrix} x' \\ y' \end{bmatrix} + \begin{bmatrix} C_1 \\ C_2 \end{bmatrix} \tag{19}$$

where $\alpha$ is the complementary angle of the slope angle and $\beta$, $C_1$, and $C_2$ are parameters related to the planting position. Equation (10) can be expressed as:

$$\begin{cases} x' = x\cos\alpha + y\sin\alpha - C_1\cos\alpha - C_2\sin\alpha \\ y' = y\cos\alpha - x\sin\alpha + C_1\sin\alpha - C_2\cos\alpha \end{cases} \tag{20}$$

According to Equations (15) and (20), the scope boundaries of the uniform distribution were obtained.

$$\begin{cases} y\cos\alpha - x\sin\alpha + C_1\sin\alpha - C_2\cos\alpha \le 1/z_r \\ y\cos\alpha - x\sin\alpha + C_1\sin\alpha - C_2\cos\alpha \ge -1/z_r \\ x\cos\alpha + y\sin\alpha - C_1\cos\alpha - C_2\sin\alpha \le 0 \\ x\cos\alpha + y\sin\alpha - C_1\cos\alpha - C_2\sin\alpha \ge -z_r \end{cases} \tag{21}$$

According to Equations (16) and (20), the scope boundaries of the triangular distribution were obtained.
when $z_r{}^2 - 2\tan\alpha > 0$,

$$\begin{cases} y\cos\alpha - x\sin\alpha + C_1\sin\alpha - C_2\cos\alpha \ge -\frac{2}{z_r{}^2}(x\cos\alpha + y\sin\alpha - C_1\cos\alpha - C_2\sin\alpha) - \frac{2}{z_r} \\ y\cos\alpha - x\sin\alpha + C_1\sin\alpha - C_2\cos\alpha \le \frac{2}{z_r{}^2}(x\cos\alpha + y\sin\alpha - C_1\cos\alpha - C_2\sin\alpha) + \frac{2}{z_r} \\ x\cos\alpha + y\sin\alpha - C_1\cos\alpha - C_2\sin\alpha \le 0 \end{cases} \tag{22}$$

when $z_r{}^2 - 2\tan\alpha \le 0$,

$$\begin{cases} y\cos\alpha - x\sin\alpha + C_1\sin\alpha - C_2\cos\alpha \ge -\frac{2}{z_r{}^2}(x\cos\alpha + y\sin\alpha - C_1\cos\alpha - C_2\sin\alpha) - \frac{2}{z_r} \\ y\cos\alpha - x\sin\alpha + C_1\sin\alpha - C_2\cos\alpha \ge \frac{2}{z_r{}^2}(x\cos\alpha + y\sin\alpha - C_1\cos\alpha - C_2\sin\alpha) + \frac{2}{z_r} \\ x\cos\alpha + y\sin\alpha - C_1\cos\alpha - C_2\sin\alpha \le 0 \end{cases} \tag{23}$$

According to Equations (17) and (20), the scope boundaries of the parabolic distribution were obtained.

$$\begin{cases} y \cos\alpha - x \sin\alpha + C_1 \sin\alpha - C_2 \cos\alpha \geq \frac{6}{z_r^3}(x \cos\alpha + y \sin\alpha - C_1 \cos\alpha - C_2 \sin\alpha)^2 \\ \qquad\qquad\qquad\qquad + \frac{6}{z_r^2}(x \cos\alpha + y \sin\alpha - C_1 \cos\alpha - C_2 \sin\alpha) \\ y \cos\alpha - x \sin\alpha + C_1 \sin\alpha - C_2 \cos\alpha \leq -\frac{6}{z_r^3}(x \cos\alpha + y \sin\alpha - C_1 \cos\alpha - C_2 \sin\alpha)^2 \\ \qquad\qquad\qquad\qquad - \frac{6}{z_r^2}(x \cos\alpha + y \sin\alpha - C_1 \cos\alpha - C_2 \sin\alpha) \end{cases} \tag{24}$$

According to Equations (18) and (20), the scope boundaries of the exponential distribution were obtained.

$$\begin{cases} y \cos\alpha - x \sin\alpha + C_1 \sin\alpha - C_2 \cos\alpha \geq \frac{e^{x \cos\alpha + y \sin\alpha - C_1 \cos\alpha - C_2 \sin\alpha} - e^{-z_r}}{z_r e^{-z_r} + e^{-z_r} - 1} \\ y \cos\alpha - x \sin\alpha + C_1 \sin\alpha - C_2 \cos\alpha \leq \frac{e^{-z_r} - e^{x \cos\alpha + y \sin\alpha - C_1 \cos\alpha - C_2 \sin\alpha}}{z_r e^{-z_r} + e^{-z_r} - 1} \\ x \cos\alpha + y \sin\alpha - C_1 \cos\alpha - C_2 \sin\alpha \leq 0 \end{cases} \tag{25}$$

$C_1$, $C_2$, and $\alpha$ in Equations (21)–(25) are discussed under three different conditions: vegetation growth on the slope's surface, upper slope region, and lower slope region. Figure 5 shows the geometric parameters of the vegetated slope (taking a uniform distribution as an example), green areas indicate the root zones.

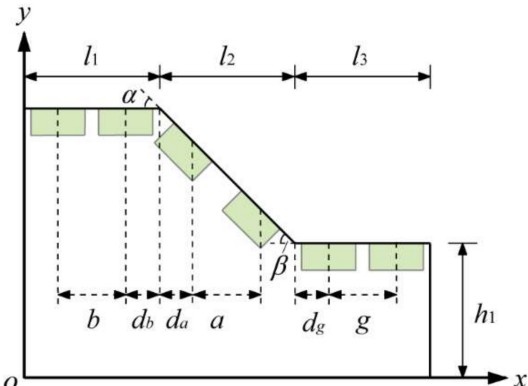

**Figure 5.** Geometric parameters of the vegetated slope.

When vegetation is planted on the slope's surface:

$$\begin{cases} \alpha = 90^\circ - \beta \\ C_{1,surface} = l_1 + d_a - a(i_{\text{tree}} - 1) \\ C_{2,surface} = h_1 + \{l_2 - [d_a + a(i_{\text{tree}} - 1)]\} \tan\beta \end{cases} \tag{26}$$

where $a$ is the horizontal interval between root centres on the slope's surface, $d_a$ is the horizontal distance between the root centre of the highest vegetation and the near edge of the upper slope region, $d_a = a/2$, $i_{\text{tree}}$ is the amount of vegetation on the slope's surface, $1 \leq i_{\text{tree}} \leq l_2/a$, and $l_2/a$ is an integer.

When vegetation grows on the upper slope region:

$$\begin{cases} \alpha = 90^\circ \\ C_{1,upper} = d_b + b(j_{\text{tree}} - 1) \\ C_{2,upper} = h_1 + l_2 \tan\beta \end{cases} \tag{27}$$

where $b$ is the horizontal interval between root centres on the upper slope region, $d_b$ is the horizontal distance between the root centre of the vegetation closest to the slope's surface and the top edge of the slope's surface, $d_b = b/2$, $j_{\text{tree}}$ is the number of vegetation on the upper slope region, $1 \leq j_{\text{tree}} \leq l_1/b$, and $l_1/b$ is an integer.

When vegetation is planted on the lower slope region:

$$\begin{cases} \alpha = 90^{\circ} \\ C_{1,lower} = l_1 + l_2 + d_g + g(k_{\text{tree}} - 1) \\ C_{2,lower} = h_1 \end{cases} \tag{28}$$

where $g$ is the horizontal interval between the root centres in the lower slope region, $d_g$ is the horizontal distance between the root centre of the vegetation closest to the slope's surface and the bottom edge of the slope's surface, $d_g = g/2$, $k_{\text{tree}}$ is the amount of vegetation in the lower slope region, $1 \leq k_{\text{tree}} \leq l_3/g$, and $l_3/g$ is an integer.

### 2.5. Numerical Implementation

The geometry and boundary conditions of the slope-stability problem are shown in Figure 6. The boundary conditions were set as rollers along the left and right vertical boundaries and were fully fixed at the base.

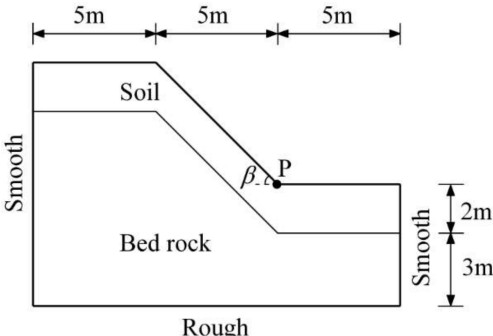

**Figure 6.** Geometry and boundary conditions of the slope-stability problem.

The soil behaviour was modelled by an elastic-perfectly plastic Mohr–Coulomb material, and the material properties are listed in Table 2. The material parameters were obtained by referring to [30,77]. The slope was divided into lower bedrock and upper soil. The root zone is the soil elements influenced by vegetation [29]. It is assumed that two adjacent root zones may overlap without affecting root distribution. Figure 7 shows root zones and overlapping root zones of different root architectures in vegetated slopes (slope angle of 45°, root depth of 1 m, root zone area of 2 m², planting distances of 2.5 m, and planting location on the slope's surface).

**Table 2.** Material properties.

| Material | Unit Weight $\gamma$ (kN/m³) | Young's Modulus E/MPa | Poisson's Ratio $v$ | Cohesion $c'$/kPa | Friction Angle $\varphi'$ (°) |
|---|---|---|---|---|---|
| Bedrock | 21 | 100 | 0.3 | 30 | 35 |
| Soil | 20 | 60 | 0.3 | 10 | 20 |
| Root zone | 20 | 60 | 0.3 | 53.2 | 20 |
| Overlapping root zone | 20 | 60 | 0.3 | 96.4 | 20 |

To guarantee both the numerical accuracy and the computational efficiency, a non-uniform initial particle distribution is assumed. A very fine mesh discretisation is used for the soil and root zones near the slope's surface, and a coarser mesh discretisation is used for the bedrock, which is situated away from the slope's surface. The meshes adopt six-node triangular elements. Table 3 shows the mesh parameters of slopes angle of 40°, 45°, and 50°.

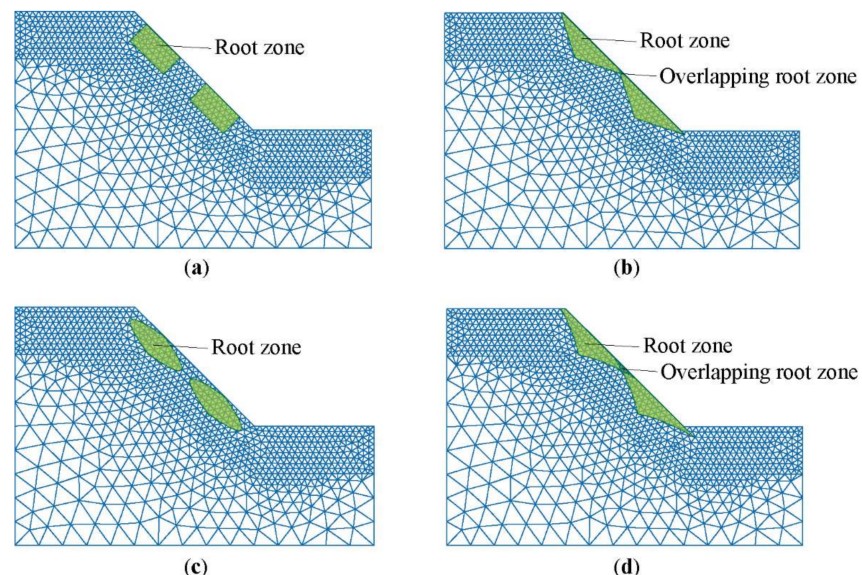

**Figure 7.** Root zones and overlapping root zones of different root architectures in vegetated slopes: (**a**) Uniform distribution; (**b**) Triangular distribution; (**c**) Parabolic distribution; (**d**) Exponential distribution.

**Table 3.** Mesh parameters.

| Slope Angle $\beta$ | 40° | 45° | 50° |
|---|---|---|---|
| Number of elements | 1695 | 1744 | 1863 |
| Number of nodes (particles) | 3498 | 3599 | 3842 |

In this study, the eSPFEM is utilised to simulate the failure process of the entire vegetated slope and predict the large deformation behaviour and final deposit of the slope failure. A kinetic energy-based criterion [32] is adopted to analyse the stability of vegetated slopes. This approach is based on the relation of the kinetic energy of the slope with the simulation time [32]. The peak value of kinetic energy curve can be considered as an indicator of the critical state, because the failure of the slope is related to large deformation, and the kinetic energy increases significantly after failure occurs. The value of SRF at the peak of the kinetic energy curve is interpreted as the FOS of the slope, and kinetic energy reaches a steady state in a short period after the peak.

The simulation is divided into two stages: first, the displacements of all of the particles are fixed, and gravity is applied to all of the particles to achieve the initial stress field; then, the particles are allowed to move, and after the initial shear failure, the unstable soil moves and reaches an equilibrium state at a new slope configuration. The horizontal displacement of point P, which is located at the slope's toe, is monitored (see Figure 5).

Slope failure was calculated using the shear strength reduction technique by gradually increasing the SRF. For different SRFs, the simulation was implemented with eSPFEM, and a physical time of 6 s [32] was considered for each SRF to obtain a new steady state for the slope after failure. To evaluate the effect of roots on slope stability, five variables (root architecture, planting distance, root depth, slope angle, and planting location) were systematically varied to calculate FOS values under different conditions.

## 3. Results

### 3.1. Comparison of the Instability of the Vegetated Slope and Bare Slope

Case 1 compares the failure mechanisms of the bare and vegetated slopes. The eSPFEM was used to analyse the stability of bare and vegetated slopes with a slope angle of 45° (planting location is the slope's surface, root depth of 1 m, root zone area of 2 m$^2$, planting

distance of 2.5 m, and a uniform root distribution) for various SRFs. Figure 8 shows the evolution of the kinetic energy with time. Figure 9 shows the variation in the maximum horizontal displacement at the slope's toe with time. Figure 10 shows the equivalent plastic strain and final configurations for various SRFs, and the grey zones indicate the initial root areas.

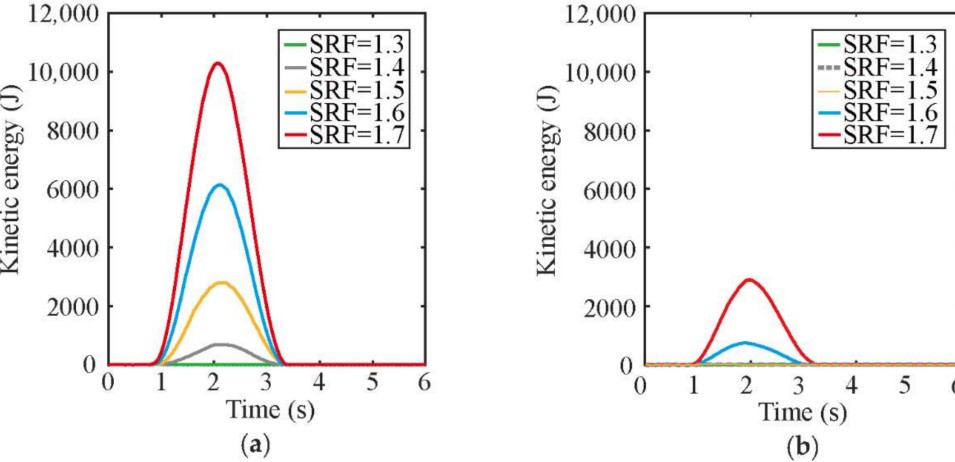

**Figure 8.** Evolution of kinetic energy with time for various SRFs: (**a**) Bare slope; (**b**) Vegetated slope.

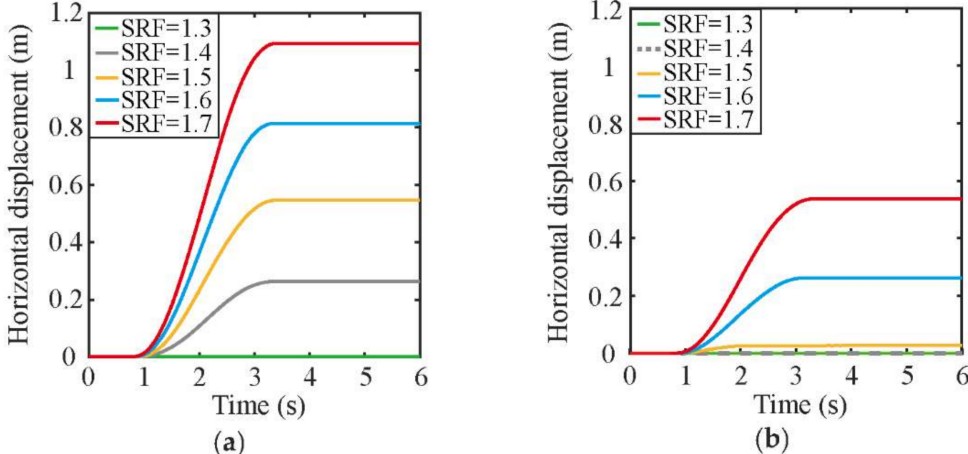

**Figure 9.** Variation of maximum horizontal displacement at the slope's toe with time for various SRFs: (**a**) Bare slope; (**b**) Vegetated slope.

For the bare slope (Figure 8a), when SRF $\leq$ 1.3, the slope is stable, and no obvious kinetic energy occurs. When SRF $\geq$ 1.4, the kinetic energy-time curve exhibits a peak value, and then the kinetic energy reaches a steady state within a short time. Therefore, the critical SRF value was 1.3, which was regarded as the FOS of the slope. Displacement at the slope's toe (Figure 9a) also occurred when SRF $\geq$ 1.4. In addition, from the results of the equivalent plastic strain (Figure 10a), it can be seen that when SRF = 1.2, a plastic strain develops at the weak band between bedrock and soil as well as at the slope's toe. When SRF = 1.3, a narrow local band of plastic strain was detected, and the plastic zone subsequently extended from the slope's toe to the slope top. The plastic strain band was dark near the top of the slope, indicating that the slope was not completely damaged. When SRF = 1.4, a continuous band of plastic strain localisation was observed, and slope failure occurred, accompanied by a large increment in displacement. The deformation of the slope was mainly concentrated on the slope's surface, and the maximum deformation occurred at the centre of the slope's surface.

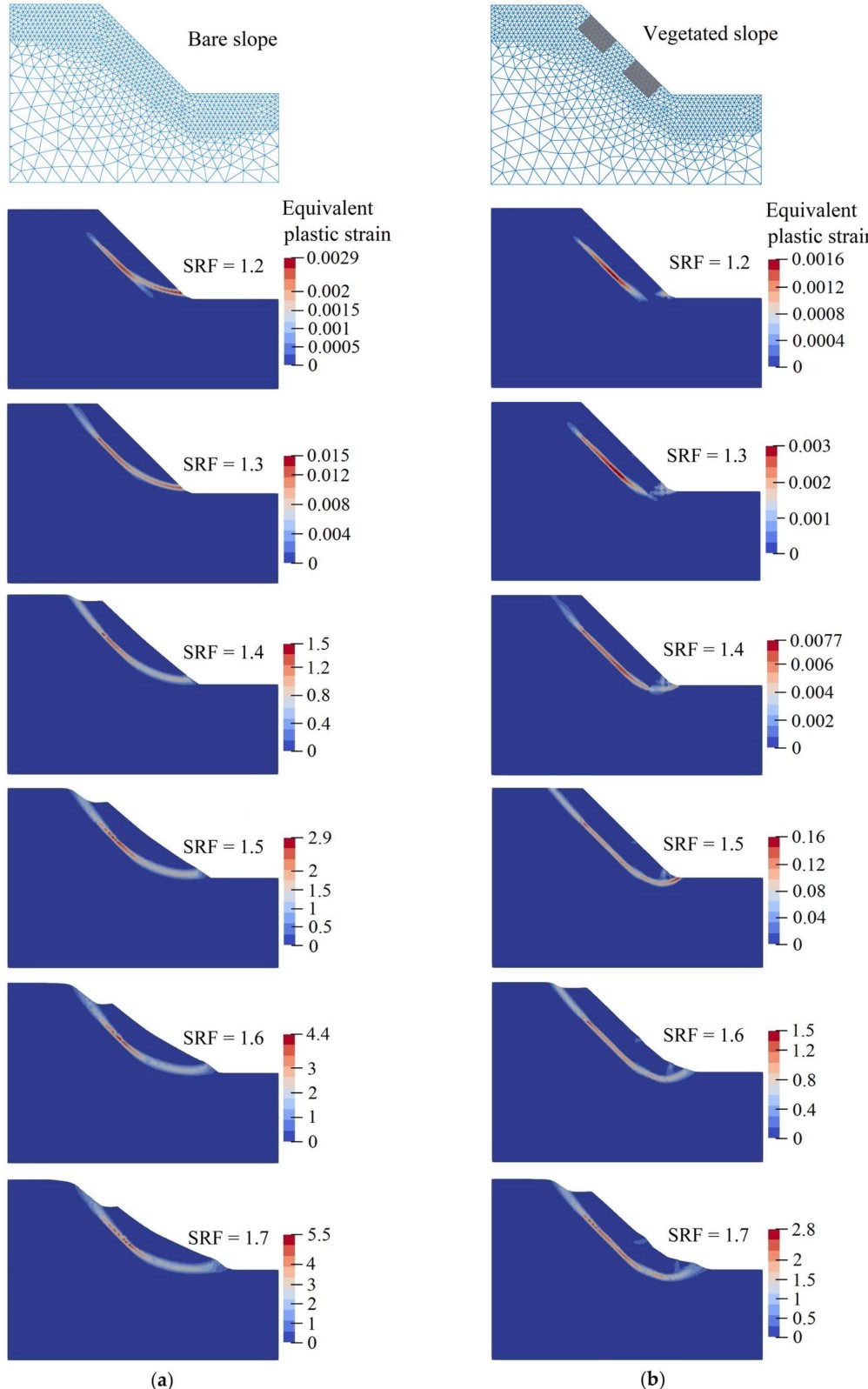

**Figure 10.** Equivalent plastic strain and final configurations for various SRFs: (**a**) Bare slope; (**b**) Vegetated slope.

For the vegetated slope (Figure 8b), no obvious kinetic energy appears when SRF $\leq 1.5$, whereas the kinetic energy-time curve shows a peak value when SRF $\geq 1.6$. The FOS of the slope was 1.5. A large deformation displacement at the slope's toe (Figure 9b) also

occurs during FOS ≥ 1.6. Moreover, according to the equivalent plastic strain diagram (Figure 10b), when SRF = 1.6, the slope produces a continuous band of plastic strain, and the upper soil mass creates a displacement. The depth of the shear band corresponded to the root zone depth.

### 3.2. Effects of the Planting Distance on the Stability of the Vegetated Slopes

The research object of case 2 was a vegetated slopes with a slope angle of 45°, root depth of 1 m, root zone area of 2 m², and planting location on the slope's surface. The effects of the root architecture on slope stability are discussed when the planting distances, *a*, are set to 5 m, 2.5 m, and 1.25 m, respectively. Figures 11 and 12 show the maximum horizontal displacement at the slope's toe after the failure of the vegetated slopes with the increase in SRF for different planting distances.

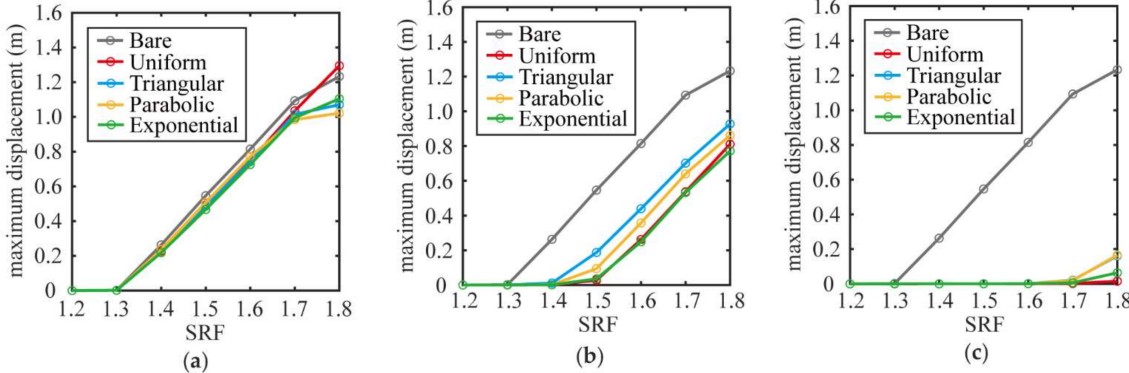

**Figure 11.** With the increase in SRF, the maximum horizontal displacement at the slope's toe for different root architectures after slope failure: (**a**) $a = 5\ m$; (**b**) $a = 2.5\ m$; (**c**) $a = 1.25\ m$.

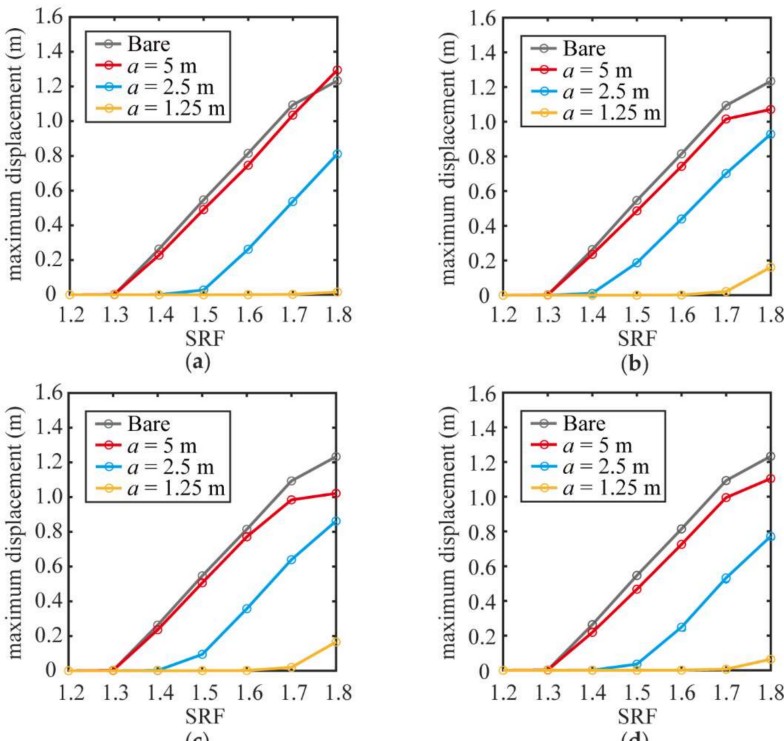

**Figure 12.** With the increase in SRF, the maximum horizontal displacement at the slope's toe for various planting distances after slope failure: (**a**) Uniform distribution; (**b**) Triangular distribution; (**c**) Parabolic distribution; (**d**) Exponential distribution.

As shown in Figures 11 and 12, when the planting distance on the slope's surface was 2.5 m or 1.5 m, the slope stability was significantly improved compared with the bare slope (Figure 11b,c). However, the effect was not evident when the distance was 5 m (Figure 11a). When the planting distance is 5 m, the slope FOS is 1.3 for the four root architectures, which is equal to that of the bare slope. When the distance is 2.5 m, the FOS of the uniform and exponential root architectures is 1.5, and that of the triangular and parabolic root architectures is 1.4. The FOS of the four root architectures was 1.7 when the distance was 1.25 m.

### 3.3. Role of the Root Depth in the Stability of the Vegetated Slopes

The study object of case 3 is a vegetated slope with a slope angle of 45°, planting distances of 2.5 m, root zone area of 2 m², and planting location on the slope's surface. The effects of root architecture on the slope stability are discussed when the root depths, $z_r$, are set to 0.5 m, 0.75 m, 1.0 m, 1.25 m, and 1.5 m, respectively. Figures 13 and 14 show the maximum horizontal displacement at the slope's toe after the failure of the vegetated slopes with an increase in SRF for various root depths.

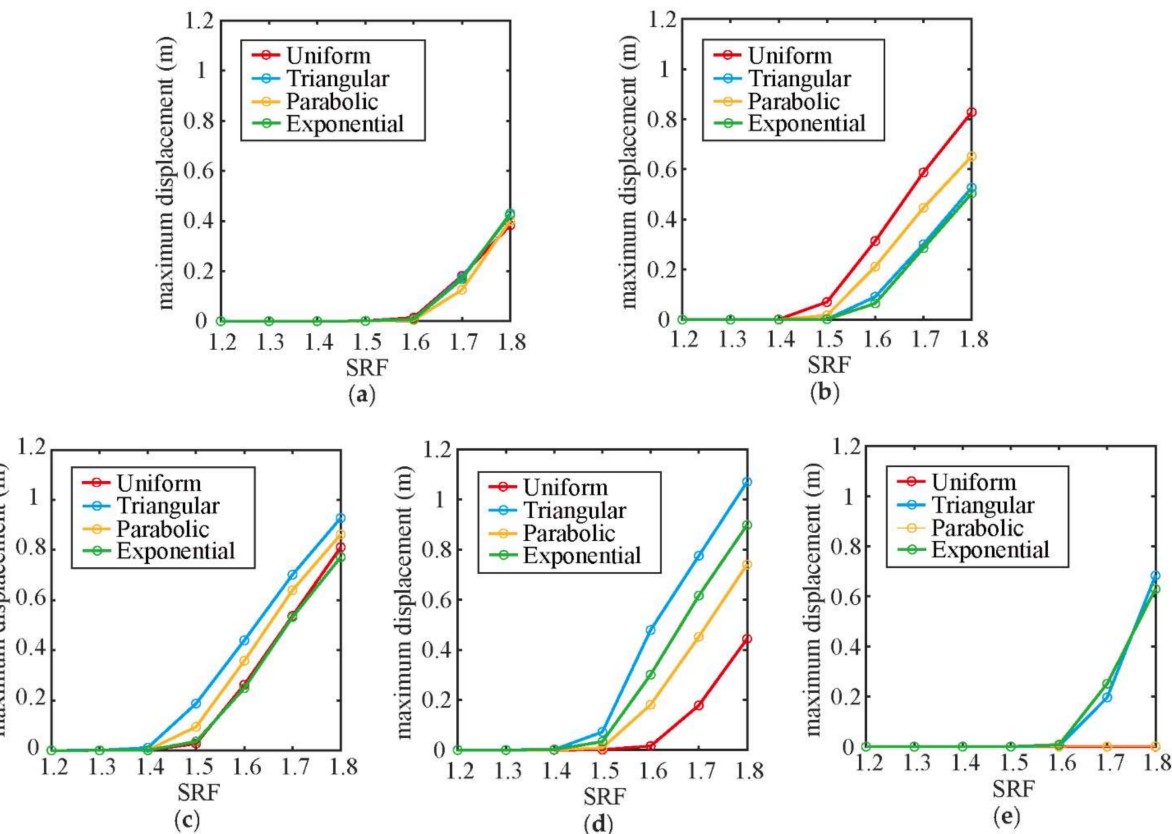

**Figure 13.** With the increase in SRF, the maximum horizontal displacement at the slope's toe for different root architectures after slope failure: (**a**) $z^r = 0.5\ m$; (**b**) $z^r = 0.75\ m$; (**c**) $z^r = 1\ m$; (**d**) $z^r = 1.25\ m$; (**e**) $z^r = 1.5\ m$.

As shown in Figure 13, after slope failure, the horizontal displacement at the slope's toe with root depths of 0.5 m and 1.5 m is smaller than that with root depths of 0.75 m, 1.0 m, and 1.25 m. When the root depth is 1.5 m, the FOS of the triangular and exponential root architectures is 1.6, whereas that of the uniform and parabolic root architectures is greater than 1.8. The FOS of the four root architectures was 1.6 when the root depth was 0.5 m.

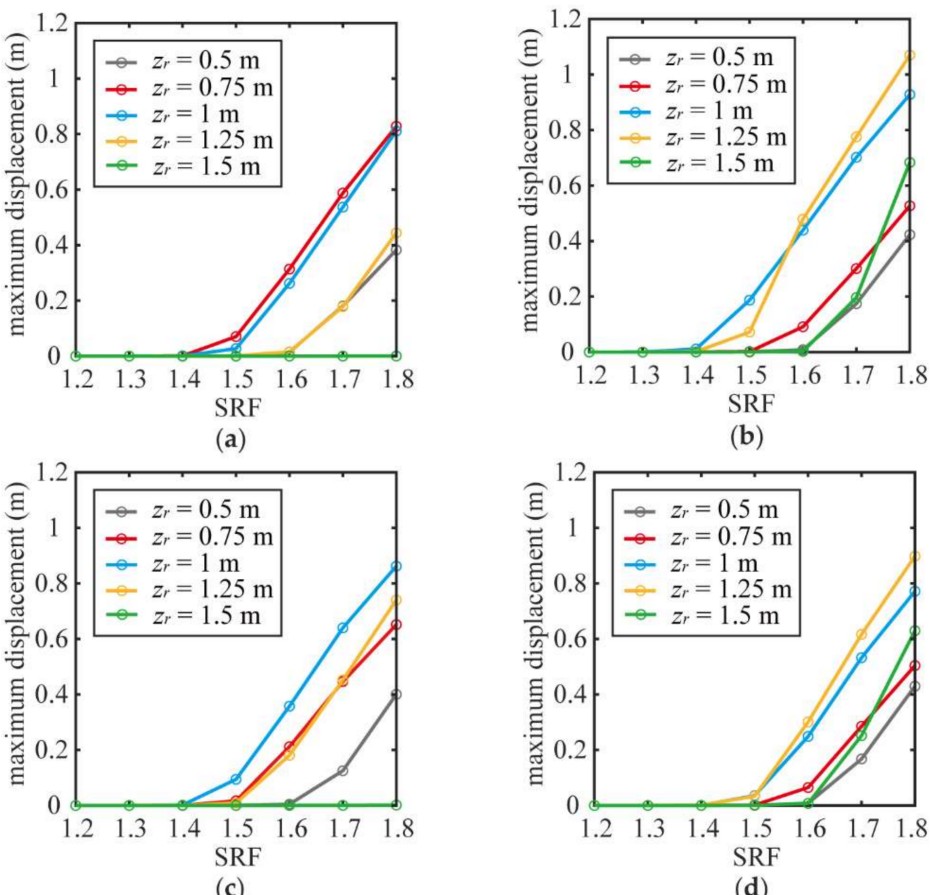

**Figure 14.** With the increase in SRF, the maximum horizontal displacement at the slope's toe for various root depths after slope failure: (**a**) Uniform distribution; (**b**) Triangular distribution; (**c**) Parabolic distribution; (**d**) Exponential distribution.

Figure 14 illustrates that, for uniform (Figure 14a) and parabolic root architectures (Figure 14c), the slope stability is the best when the root depth is 1.5 m (FOS > 1.8). When the root depth of the uniform root architecture was 0.75 m and that of the parabolic root architecture was 1.0 m, the slope stability was the worst (FOS = 1.4). For triangular (Figure 14b) and exponential (Figure 14d) root architectures, the FOS reaches the maximum when root depths are 0.5 m and 1.5 m (FOS = 1.6); and the FOS is the lowest when the root depths are 1.0 m and 1.25 m (FOS = 1.4).

### 3.4. Effects of the Slope Angle on the Stability of the Vegetated Slopes

The research object of case 4 is a vegetated slope with a planting distance of 2.5 m, root depth of 1 m, root zone area of 2 m$^2$, and planting location is the slope's surface. The influence of root architecture on slope stability is discussed when the slope angles, $\beta$, are set to 40°, 45°, and 50°, respectively. Figures 15 and 16 show the maximum horizontal displacement at the slope's toe after the failure of vegetated slopes with an increase in the SRF for different slope angles.

As shown in Figures 15 and 16, when the slope angles are 40°, 45°, and 50°, the FOS are 1.3, 1.5, and 1.7, respectively, for the uniform and exponential root architectures; for triangular root architecture, the FOS are 1.2, 1.4, and 1.6, and for parabolic root architecture, the FOS are 1.3, 1.4, and 1.6, respectively.

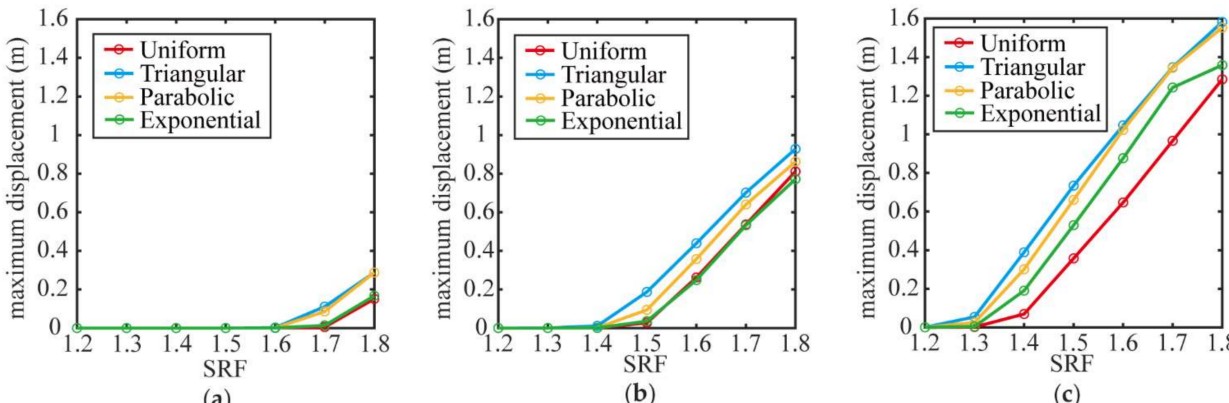

**Figure 15.** With the increase in SRF, the maximum horizontal displacement at the slope's toe for different root architectures after slope failure: (**a**) $\beta = 40°$; (**b**) $\beta = 45°$ (**c**) $\beta = 50°$.

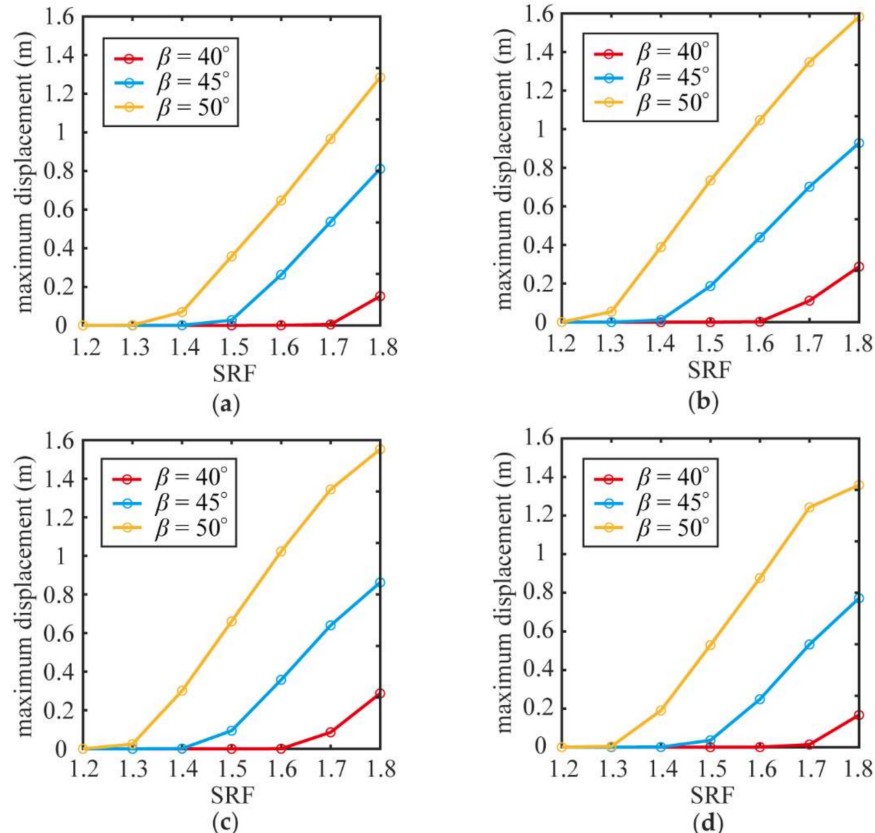

**Figure 16.** With the increase in SRF, the maximum horizontal displacement at the slope's toe for various slope angles after slope failure: (**a**) Uniform distribution; (**b**) Triangular distribution; (**c**) Parabolic distribution; (**d**) Exponential distribution.

### 3.5. Influence of the Planting Location on the Stability of the Vegetated Slopes

The study object of case 5 is a vegetated slope with a slope angle of $45°$, planting distance of 2.5 m, root depth of 1 m, and root zone area of 2 m$^2$. The effects of the root architecture on slope stability when the location of the root zone changes are discussed. We referred to Chok et al. [29] for the planting locations. Figure 17 shows the slopes with the planting positions of the slope's surface, slope's toe, the slope's surface and toe, upper slope region, lower slope region, upper and lower slope regions, and entire ground surface, as well as their final equivalent plastic strains during FOS = 1.6. The grey area represents

the root zone (taking uniform root architecture as an example). Figures 18 and 19 show the maximum horizontal displacement at the slope's toe after slope instability with an increase in SRF for different planting locations.

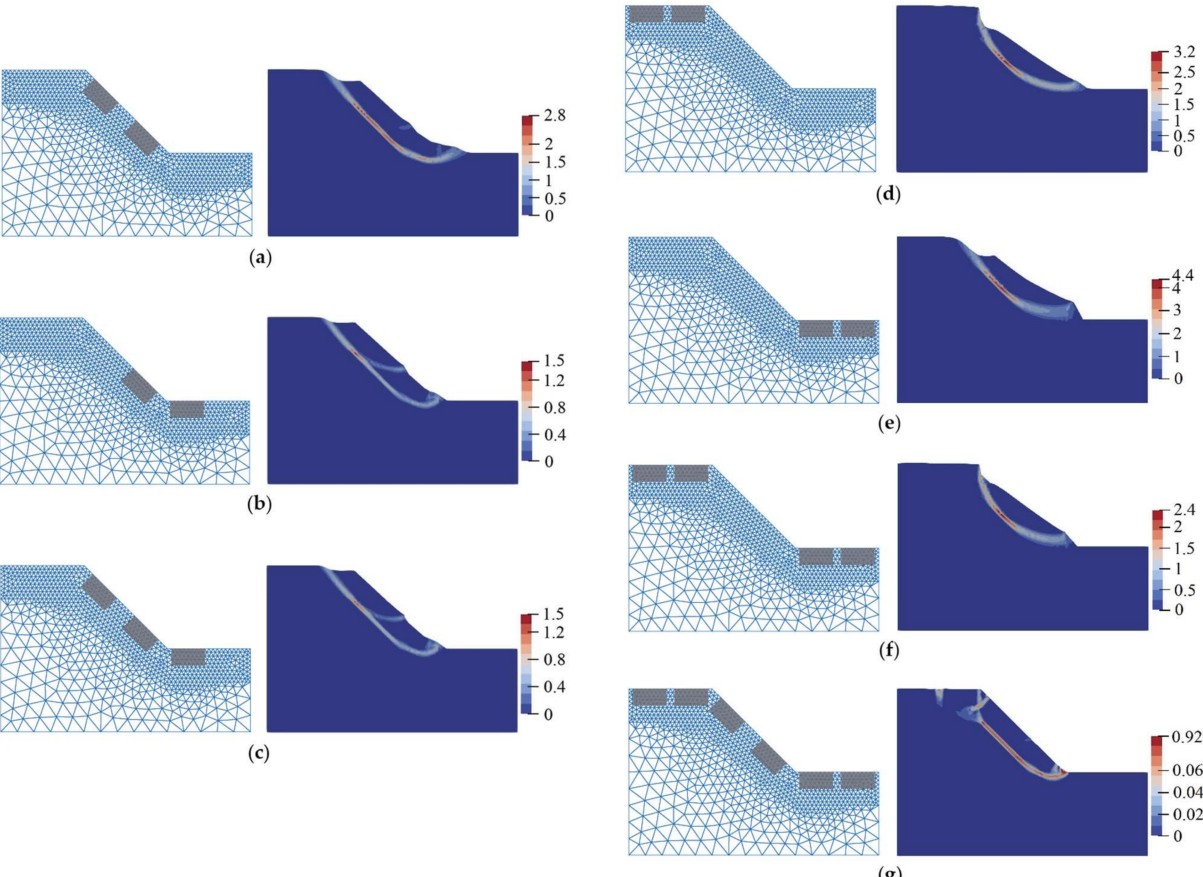

**Figure 17.** Vegetated slopes for different planting locations and the final equivalent plastic strain for FOS = 1.6: (**a**) slope's surface; (**b**) slope's toe; (**c**) slope's surface and toe; (**d**) upper slope region; (**e**) lower slope region; (**f**) upper and lower slope regions; (**g**) entire ground surface.

Figure 17 shows that different planting positions have different influences on the shear band of the slope failure. The improvement in slope stability was the most obvious in the entire ground surface planting. Planting at the slope's toe can effectively reduce the sliding displacement of the slope soil.

As shown in Figure 16, vegetation on the entire ground surface (Figure 18g) had the best effect on the slope stability (FOS > 1.8). Planting on the lower slope region (Figure 18e) has little impact on the FOS, which is equal to that of the bare slope (FOS = 1.3). Planting on the slope's surface (Figure 18a) was better than that on the lower slope region, and the influence of uniform and exponential root architectures (FOS = 1.5) was better than that of triangular and parabolic root architectures (FOS = 1.4). The effects of planting on the slope's toe (Figure 18b) and the slope's surface and toe (Figure 18c) were both similar. Both were better than the slope's surface, and the impacts of the triangular and exponential root architectures (FOS = 1.7) were better than those of uniform and parabolic root architectures (FOS = 1.5). The effects of planting on the upper slope region (Figure 18d) and upper and lower slope regions (Figure 18f) were similar, and the influence of uniform architecture (FOS > 1.8) was better than that of the other three root architectures (FOS = 1.5).

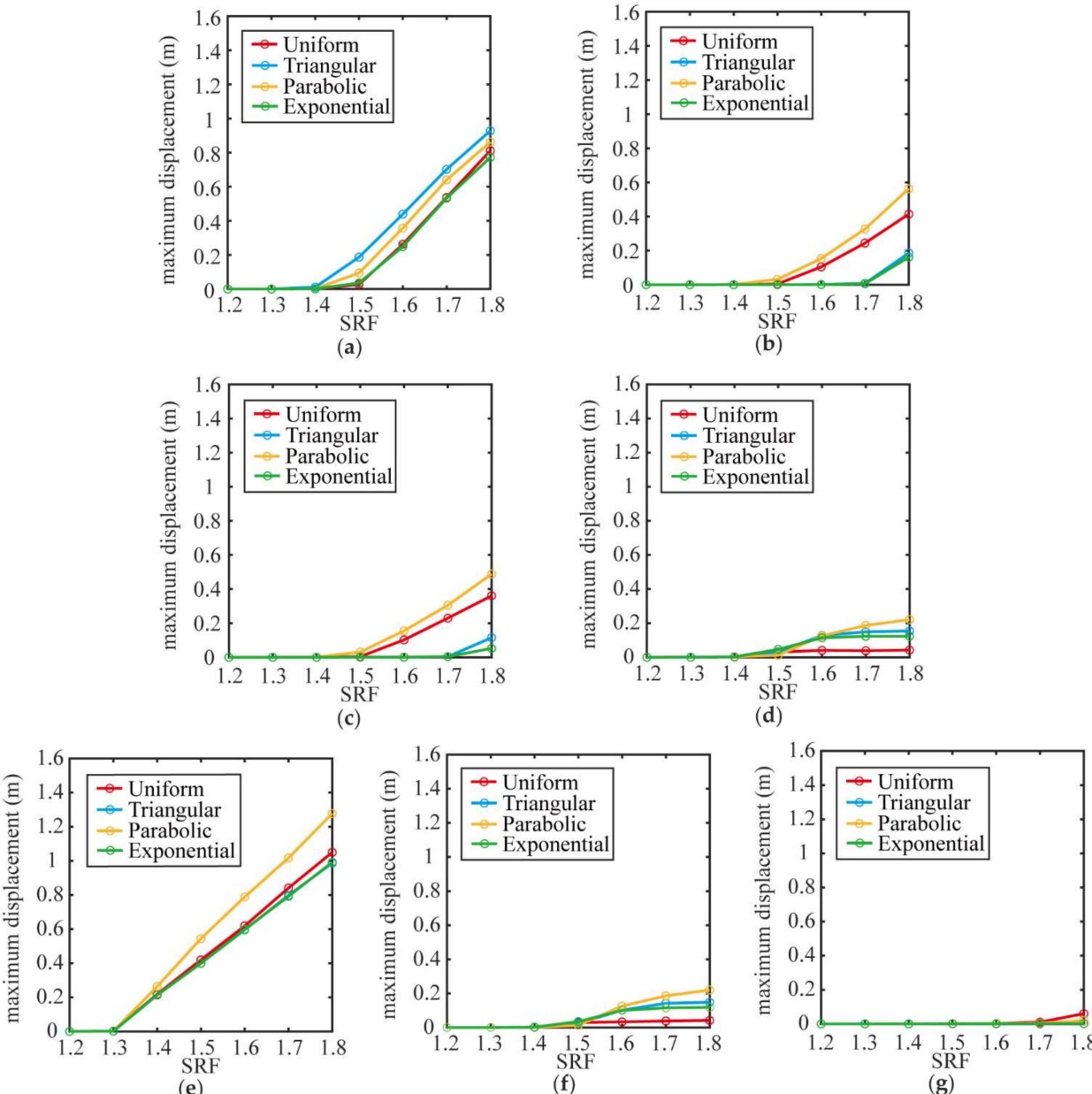

**Figure 18.** With the increase in SRF, the maximum horizontal displacement at the slope's toe for different root architectures after slope failure: (**a**) slope's surface; (**b**) slope's toe; (**c**) slope's surface and toe; (**d**) upper slope region; (**e**) lower slope region; (**f**) upper and lower slope regions; (**g**) entire ground surface.

In Figure 19, the planting of uniform root architecture (Figure 19a) on the upper and lower slope regions, upper slope region, and entire ground surface (FOS > 1.8) are better than those on the slope's surface and toe, as well as on the slope's surface (FOS = 1.5). The planting effects of parabolic root architecture (Figure 19c) on the upper and lower slope regions, slope's surface, and upper slope region were weaker than those of the uniform root architecture. The effects of triangular (Figure 19b) and exponential root architectures (Figure 19d) are similar, and they have a greater advantage when they are present at the slope's toe, as well as the slope's surface and toe (FOS = 1.7).

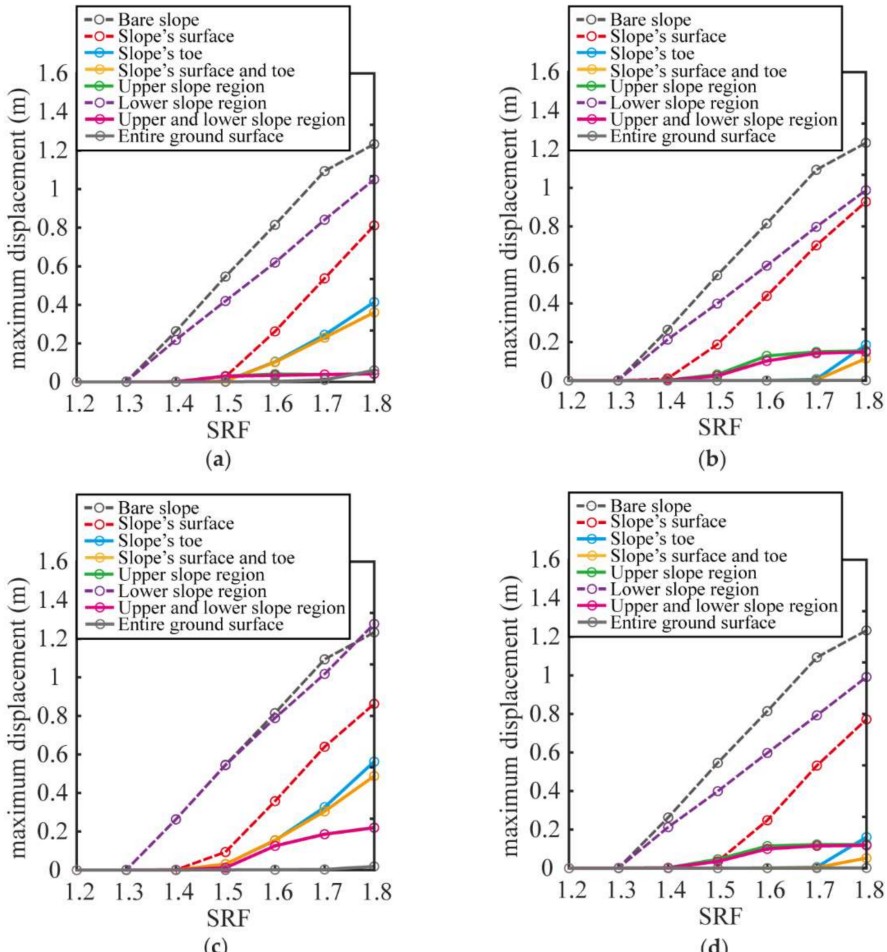

**Figure 19.** With the increase in SRF, the maximum horizontal displacement at the slope's toe of the slopes with different planting locations after slope failure: (**a**) Uniform distribution; (**b**) Triangular distribution; (**c**) Parabolic distribution; (**d**) Exponential distribution.

## 4. Discussion

In this study, the slip surface of the bare slope was mainly distributed in the weak layer at the interface of the rock and soil, whereas the slip surface of the vegetated slope was distributed at the bottom of the root zone, and the upper layer of soil entrains the root system to slip. The tensile strength and adhesion properties of the roots reinforced the soil. Plant roots with high tensile strength increase the confining stress of the soil through their compact root matrix system [29]. The results of the finite element analysis showed that reinforcement of the root can improve the stability of the slope, reduce the displacement of the landslide, and increase the FOS value.

The FOS value of the vegetated slope increased with decreasing planting distance. The higher the planting density, the stronger the root-strengthening effect. It should be noted that if the planting distance exceeds a certain range, then the slope stability may not improve. In this study, when the planting distance on the slope was 5 m, the effect was similar to that on the bare slope. The stability of a slope with a uniform root architecture is most sensitive to the planting distance.

For many slopes, the root depth is usually limited by bedrock, which is usually shallow and less than 2 m [58]. The failure depth of most slopes is between 0.5 m and 1 m, and the root zone plays a role in mechanical stability only when the root depth reaches the deeper soil layer [78]. With an increase in root depth, the FOS first decreased and then increased. When the root depth is 0.5 m, the slope's surface is covered with overlapping roots, which is similar to the geosynthetic reinforcement on the slope's surface. When the root depth

was 1.5 m, it was similar to installing anti-slide piles or anchors. In addition, for uniform and parabolic root architectures, a greater root depth is more beneficial for slope stability. For triangular and exponential root architectures, shallow roots were more conducive. In addition, the stability of a slope with a uniform and parabolic root architecture is more sensitive to its root depth.

The slope FOS of the four root architectures decreased with an increase in slope angle. Regardless of whether the slope is steep or has a slight incline, uniform and exponential root architectures are more effective in improving the slope stability.

The position of the plants on the slope also affects their contribution to stability. Except for planting in the lower slope region, the stability of the vegetated slope in the other planting positions was better than that of the bare slope. If there is no restriction on planting location and vegetation quantity, planting on the entire ground surface of a slope has the best effect on improving slope stability, which is similar to the conclusion of [29]. Moreover, it is better to plant vegetation with a uniform root architecture in the upper slope region or plant vegetation with a triangular or exponential root architecture on the slope's toe. When the vegetated slope is unstable, the depth of the plastic strain at the top and toe of the slope is shallower; therefore, it is more conducive for roots to play a role. Tensile cracks may occur at the top of a slope, and the roots provide traction and support soil to prevent landslides [79]. The roots bear pressure at the toe of the slope, which can act as a support and inhibit soil sliding [80].

## 5. Conclusions

In this study, the eSPFEM was utilised to simulate the instability of vegetated slopes with large deformations. This method can reasonably predict the deformation process of the slope structure and the final deposition, avoiding the difficulty of numerical calculations and loss of calculation accuracy.

The Mohr–Coulomb constitutive model was extended by introducing the additional soil cohesion, $c_r$, generated by roots and therefore increasing the shear strength of the soil. The boundary functions of the four root architectures (uniform, triangular, parabolic, and exponential) on the slope were derived. The FOS values of the four root architectures for various planting distances, root depths, slope angles, and planting locations were calculated using the shear strength reduction technique with a kinetic energy-based criterion, and the effects of root architecture on slope stability were evaluated.

Their results showed that roots can effectively improve slope stability and reduce landslide displacement. The higher the planting density, the stronger the root-strengthening effect. With an increase in root depth, the FOS first decreased and then increased. For uniform and parabolic root architectures, deeper roots are more beneficial to slope stability, whereas for triangular and exponential root architectures, shallower roots are more conducive. The FOS decreases with the increase in slope angle; uniform and exponential root architectures are more effective in improving slope stability, no matter whether the slope is steep or has a slight incline. Vegetation at the slope's toe can effectively reduce the sliding displacement of the slope soil. Planting on the entire ground surface had the best effect on slope stability improvement, followed by planting vegetation with uniform root architecture in the upper slope region or planting vegetation with triangular or exponential root architectures on the slope's toe.

This study provides valuable information on the contribution of different root architectures to slope stability and can guide the selection of vegetation species and planting locations, which can contribute to improving slope stability and optimising the management of mountain shelter forests. Nonetheless, the limitation of this model lies in the need for accurate parameterisation and a large amount of calculation, so it is currently only applicable to small slopes.

**Author Contributions:** Conceptualization, X.J., W.Z. and P.C.; methodology, X.J. and W.Z.; software, Y.J. and W.Z.; validation, X.J. and X.W.; data curation, X.J. and Y.J.; writing—original draft preparation, X.J.; writing—review and editing, X.J., Y.J. and P.C.; visualization, X.J. and X.W.; funding acquisition, Y.J. and P.C. All authors have read and agreed to the published version of the manuscript.

**Funding:** This research was funded by the Science and Technology Program of Guangzhou, grant numbers 201605030009 and 201803020036, the National Natural Science Foundation of China, grant number 42101422, Effect of changes of underlying surface on precipitation based on radar and optical remote sensing: A case study of Guangzhou City, grant number 202102021287, Research Project on Key Technologies of Watershed Water Governance, grant number 2021-09, the characteristic innovation project of Guangdong Provincial Department of Education, grant number 2022KTSCX013, Guangdong Engineering and Research Center for Unmanned Aerial Vehicle Remote Sensing of Agricultural Water and Soil Information, and Shenzhen Smart Water Project Phase I—Soil and water conservation Information construction project.

**Institutional Review Board Statement:** Not applicable.

**Informed Consent Statement:** Not applicable.

**Data Availability Statement:** The data presented in this study are available upon request from the corresponding author.

**Conflicts of Interest:** The authors declare no conflict of interest.

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
