# Peer review of "Numerical Analysis of an Explicit Smoothed Particle Finite Element Method on Shallow Vegetated Slope Stability with Different Root Architectures"

_sustainability, doi:10.3390/su141811272_

Round 1
Reviewer 1 Report
This paper deals with a very interesting topic, an explicit smoothed particle finite element method was introduced to solve the stability problems of the shallow vegetated slopes. Some questions are listed as follows:
1. The formulations of the explicit smoothed particle finite element method used in the work were not given to the audience. It is very helpful for readers who do not familiar with the method to figure out the entire process of the calculation.
2. How is the grid of the root zone discrete and is the element size consistent with that of the soil region? Is it necessary to define in advance which particles belong to the root zone before the calculation? How do we know if the root zones overlap? These questions need to be clarified before the calculation (e.g., in Section 2).
3. In the paragraph before Section 3, the authors claim the physical time of the process is 6s. In general, soil slippage could be a slow process. The authors should provide some basis or reference to show whether such a short time is appropriate.
4. In Section 3, the results of the simulation of many kinds of situations are given. However, how did the authors know these results are correct? It would be better if the authors could add some experiment results or compare their results with other research.

Reviewer 2 Report
In this paper, the authors used the eSPFEM method to simulate large deformation and analyze stability of the vegetation slope. The effects of four types of roots at different planting distances, depths, slope angles and planting locations on slope stability were analyzed by the shear strength reduction technique based on kinetic energy criterion, which provided important guidance for the selection of vegetation types and planting locations in slope management. The topic of the study is interesting and would be of interest to the readership of the journal.
1. Line 80, I think the PFEM is not belonged to the mesh-free technique. So the logic between two sentence is not direct. Please correct it.
2. In the introduction, the literature review of PFEM related studies should be presented, especially for the smoothed PFEM, such as the edge-based smoothed PFEM, node-based smoothed PFEM, or even cell-based PFEM, some recently studies should be mentioned, like “An edge-based strain smoothing particle finite element method for large deformation problems in geotechnical engineering. Int J Numer Anal Methods Geomech 44 (7):923-941. doi:10.1002/nag.3016”; “Two-phase PFEM with stable nodal integration for large deformation hydromechanical coupled geotechnical problems. Comput Methods Appl Mech Eng 392:114660”; “A stable node-based smoothed PFEM for solving geotechnical large deformation 2D problems. Comput Methods Appl Mech Eng 387:114179. doi:https://doi.org/10.1016/j.cma.2021.114179”; “A novel coupled NS-PFEM with stable nodal integration and polynomial pressure projection for geotechnical problems. Int J Numer Anal Methods Geomech n/a (n/a). doi:https://doi.org/10.1002/nag.3417”
3. The methods related to large deformation simulation should be also reviewed, and the papers in the special issue of large deformation analysis https://link.springer.com/journal/11582/volumes-and-issues/22-11 can be as a reference.
4. According to GR LIU, the NS-PFEM or NS-FEM has a softer property, so what’s the effect on the slope stability? Please comment on it.
5. The Mohr-Coulomb model is extended to simulate the effect of root by introducing an additional cohesion. Large friction angles (20~35) is used in the paper, but the extended Mohr-Coulomb model does not consider the variation of friction angle. Does the root in soil has a significant effect on the internal friction angle of soil? The authors need to explain this briefly. And in the Eq.(4), please define the cr.
6. It is better to plot the root area of vegetation in Fig. 8(b), so that readers can intuitively understand it.
7. In the Table 1, the density of soil 20 is too large for the saturated soil. The density of saturated soil is normally between 16-19 KN/m3.
Reviewer 3 Report
The Authors have explored the interesting topic of slope stabilization through vegetation. eSPFEM is used to model the slope, and the effects of various parameters such as root architecture, root depth, planting distance and slope angle on the slope stability are analyzed. The manuscript is generally well-written and presents some interesting findings. I believe it can be published in your journal after addressing some issues:
1- The introduction section could definitely be better organized. For instance, the third paragraph (lines 45-51) can be placed before the second paragraph or combined with the first paragraph to explain the importance of vegetation as a remediation technique. Some previous works are mentioned in the second paragraph, including a few FEM studies, and then in the fourth paragraph, some other FEM studies are discussed. I think they need to be combined. It would be much better to split the relevant studies into two groups of numerical and experimental studies, where a summary of their findings to date are presented. In addition, none of the studies on slope stability using meshless techniques, such as DEM, SPH and MPM, is mentioned here. This is especially important considering the particle-based method used in this study.
2- It is essential to provide at least a brief description of eSPFEM with some of the main equations (and probably a schematic showing different components). Also, did the Authors use a user-written code or a commercial software for the analysis? It must be clarified.
3- The Authors have not presented any validation cases for their proposed approach. Obviously, performing a comparison with an experimental study would be ideal, but even comparing the results (at least for the bare slope) with one of the mainstream numerical techniques or available analytical solutions can considerably increase the scientific value of the manuscript.
4- The advantages of this method over the traditional FEM are explained, but how is it compared to the other meshless techniques? Is it computationally less demanding? Is it more accurate? Or is it more stable? If there are specific benefits to it, they must be discussed.
5- What does the colorbars show in figure 8? Please add a title.
6- In Figure 7, the slope deformation stops at almost the same time (around 3 seconds) for all SRF values, which is a bit odd to me. Also, the maximum slope displacement almost linearly increases with SRF for the majority of cases (for example Figure 9). Are there any experimental studies confirming these patterns?
7- I believe section 4 (Discussion) should be merged with section 5 (Conclusions), presenting the main findings of the paper.
Reviewer 4 Report
Dear authors,
Here i attached the review comments document kindly check

Round 2
Reviewer 1 Report
It could be accepted in present form.
Reviewer 3 Report
In my opinion, the quality of the paper has been significantly improved and it's ready to be published.
Reviewer 4 Report
Dear Authors
Thank you for your revised paper. I have carefully read the revised paper and found it has been improved by responding to and clarifying the queries raised by the reviewers.
Thank you